



# Quantitative climate reconstruction from sedimentary ancient DNA: framework, validation and application

Ulrike Herzschuh[1,2,3#], Thomas Böhmer[1#], Weihan Jia[1], Simeon Lisovski[1]

[1]Polar Terrestrial Environmental Systems, Alfred Wegener Institute Helmholtz Centre for Polar and Marine Research, Telegrafenberg A45, 14473 Potsdam, Germany

[2]Institute of Environmental Science and Geography, University of Potsdam, Karl-Liebknecht-Str. 24–25, 14476 Potsdam, Germany

[3]Institute of Biochemistry and Biology, University of Potsdam, Karl-Liebknecht-Str. 24–25, 14476 Potsdam, Germany

# equal contribution

*Correspondence*: Ulrike Herzschuh (ulrike.herzschuh@awi.de)

**Abstract.** Quantitative reconstructions of terrestrial climate conditions typically rely on biological proxies such as pollen. Despite their widespread use, these proxies exhibit inherent limitations such as low taxonomic resolution and complex taphonomies. Sedimentary ancient DNA (sedaDNA), particularly plant metabarcoding using chloroplast markers (*trn*L-*gh*), has emerged as a promising alternative offering enhanced taxonomic precision and local origin. Here, we present the framework for quantitative reconstruction of summer temperatures from sedaDNA assemblages applying methods that rely on surface samples for calibration (weighted-averaging partial least squares (WA-PLS), modern analog technique (MAT)) and introducing a framework that combines modern plant occurrences and species distribution modeling (SDM) to derive taxon-specific probability density functions (PDFs) for calibration. Applying these approaches to sedaDNA data from 203 lake sediment-surface samples across Siberia, we obtained highly accurate reconstructions with median biases as low as 0.5°C and a strong correlation with observed temperatures. Our method shows a low reconstruction bias when compared to those from other proxy calibration studies. Applied to a Lake Billyakh sediment core in eastern Siberia, our sedaDNA-based reconstructions using various approaches show similar trends and successfully reproduce regional climate changes over the past 32,000 years, aligning closely with independent pollen-based records. We also reveal that higher taxonomic resolution results in a more precise reconstruction due to narrower tolerance ranges with higher taxonomic resolution. The demonstrated reliability, low bias, and superior taxonomic resolution underscore the significant potential of sedaDNA as a robust and sensitive new terrestrial proxy for quantitative paleoclimatic research.

## 1 Introduction

Quantitative climate reconstructions are essential for validating climate model performance when simulating past climate changes and thus indicating their potential to predict future climate conditions (Schmidt et al., 2014). Fossil assemblages derived from marine, lacustrine, or terrestrial archives are among the most widely used proxies providing direct quantitative





information on climate (Birks et al., 2010). These approaches build upon the assumption that compositional changes within biotic communities are predominantly driven by changing climatic conditions (Birks et al., 2010). However, all proxy-based reconstructions inherently contain uncertainties, due to ecological complexities, a lack of understanding of taphonomies, and

methodological limitations (Juggins, 2013). Although multi-proxy reconstructions are frequently employed for climate model evaluation, discrepancies between proxy data and model simulations – exemplified by the "Holocene conundrum" – are often attributed to proxy biases rather than modeling errors (Liu et al., 2014). This underscores the pressing need to develop and refine new and robust climate proxies.

Among terrestrial proxies, pollen analysis remains the most commonly used method for reconstructing past climate conditions

on land (Herzschuh et al., 2023a), largely due to its widespread preservation across various terrestrial archives such as lake sediments, peat bogs, and soil deposits (Birks and Seppä, 2004). The popularity of pollen-based reconstructions stems from the strong and extensively studied quantitative relationships linking climate variables (e.g., temperature and precipitation) with vegetation composition and pollen assemblages (e.g. Seppä et al., 2004; Herzschuh et al., 2010). Despite these strengths, pollen proxies are subject to several significant limitations (Chevalier et al., 2020). A critical issue is their limited taxonomic

resolution, typically constrained to genus or family levels, which can obscure precise ecological interpretations, particularly when closely related taxa differ in climate preferences (Tian et al., 2017). Moreover, pollen productivity is inherently biased toward wind-pollinated taxa, thereby skewing reconstructions toward certain climate conditions. For example, the contribution from productive boreal forest taxa to tundra pollen spectra leads to a warm bias (Klemm et al., 2013). Additionally, pollen signals commonly integrate information over larger spatial scales – ranging from local to regional – depending heavily on

basin size, landscape openness, vegetation composition, and transport mechanisms (Prentice, 1985; Sugita, 2007). Plant macrofossils offer valuable complementary data, as they provide higher spatial precision, typically representing local-scale vegetation dynamics and facilitate species-level identification, thereby overcoming the coarse taxonomic resolution inherent in pollen analysis (Birks, 2001). However, plant macrofossil records often have limited availability and preservation potential, resulting in fragmented or discontinuous fossil sequences that constrain the reliability and continuity of reconstructions. These

limitations emphasize the necessity of developing additional vegetation proxies capable of providing continuous, local, and quantitatively robust climate reconstructions.

Sedimentary ancient DNA (sedaDNA) has recently emerged as a novel biological proxy that effectively traces past changes in ecological communities, offering significant potential for paleoclimatic reconstructions (Parducci et al., 2017; Capo et al., 2021). Like plant macrofossils, sedaDNA primarily originates from the vegetative parts of local and nearby plants (Niemeyer

et al., 2017; Alsos et al., 2018). Yet, similar to pollen analysis, recorded DNA reads are abundant and can yield quantitative data about diverse plant taxa simultaneously. Plant metabarcoding using chloroplast DNA markers, such as the commonly employed *trn*L-*gh* fragment, enables reconstruction of past vegetation assemblages at high taxonomic resolution, mostly reaching species or genus level (Taberlet et al., 2007). Numerous sedaDNA datasets have already been published from various regions worldwide, especially from the extratropical Northern Hemisphere, demonstrating the method's broad-scale

applicability (Liu et al., 2021; Alsos et al., 2022; Courtin et al, 2025). Moreover, plant metabarcoding studies of modern lake





surface-sediments using this marker have provided promising reference datasets (Alsos et al., 2016; Niemeyer et al., 2017; Stoof-Leichsenring et al., 2020; Li et al., 2021), successfully establishing links between contemporary vegetation assemblages and subfossil sedimentary DNA profiles, thereby facilitating accurate reconstructions of past vegetation types from sedaDNA records (Liu et al., 2021). Also, first approaches to leverage information about median climatic conditions of selected taxa as

climate indicators yielded promising results (Mayfield et al., 2024). Nevertheless, despite these encouraging developments, the reliability and robustness of plant sedaDNA as a climate proxy have not yet been systematically explored or validated.

Essentially, two main approaches exist for calibrating biological assemblage proxies. The first approach directly links modern sub-fossil assemblages with contemporary climate parameters. This is done either by directly matching modern assemblages to fossil spectra using the Modern Analogue Technique (MAT; Overpeck et al., 1985), or by applying modern taxa-climate

relationships to fossil spectra using statistical methods such as Weighted-Averaging Partial Least Squares regression (WA-PLS; ter Braak and Juggins, 1993). The second approach leverages ecologically meaningful relationships by relating the occurrence of individual taxa to modern climate conditions using, for example, probability density functions (PDFs; Kühl et al., 2002), which represent the probability of occurrence across climate gradients. The individual PDFs of recorded taxa in fossil assemblages are then combined to form joint PDFs, enabling quantitative reconstruction of past climates. This PDF-

based method, initially developed for plant macrofossils due to their high taxonomic resolution, has recently been advanced through the Climate REconstruction SofTware (CREST) framework allowing, among others, for weighting by abundances (Chevalier et al., 2014). CREST also provides detailed diagnostic tools to assess the climate niches of individual taxa and quantifies their contributions to climate reconstructions at the sample level. A significant advantage of calibrating proxies using modern plant occurrences is that it does not depend on the availability of an extensive modern surface-sample set for

training. If, however, modern sub-fossil assemblages are available, they can be employed to independently validate this method, thereby enhancing confidence in the resulting proxy-based climate reconstructions. Hence, despite the availability of suitable statistical frameworks, their assessment and application to sedaDNA data have not yet been tested.

Although large global databases of taxonomic occurrences, such as the Global Biodiversity Information Facility (GBIF; GBIF.org, 2025), provide extensive data on species distributions, establishing accurate regional PDFs requires comprehensive

modern occurrence data specifically from the target study regions (Meyer et al., 2016). This presents a substantial challenge, especially in remote and data-scarce regions such as the Arctic and Siberia. To address gaps in observational data, species distribution models (SDMs) have increasingly been employed to predict species occurrences based on their environmental niche preferences and thus allow for continuous information about taxa occurrences along the variable gradient (Elith and Leathwick, 2009; Thuiller et al., 2009). Recent methodological advances have enabled the efficient implementation of SDMs

for hundreds of species simultaneously and such allowing for large-scale biodiversity approaches (Hu et al., 2025). However, despite their extensive use in contemporary ecological and biogeographic studies, SDMs have not yet been systematically integrated into quantitative paleoclimate reconstructions.

Here, we present a framework for leveraging plant metabarcoding of sedaDNA as a proxy for quantitative reconstruction of summer temperatures. The framework uses PDFs derived from modern relationships between 180 plant taxa occurrences



(representing metabarcoding sequence types) and summer temperature, established using newly developed SDMs. We validated this approach using plant metabarcoding data from 203 lake surface-sediment samples across Siberia, and subsequently applied it to reconstruct summer temperature changes at a test site in eastern Siberia within the region. We also evaluated this method against methods employing modern surface sediments for calibration. We validate the calibration method and reconstruction using pollen-based calibration and reconstruction.

## 2 Materials and methods


**Figure 1: General framework illustrating the use of sedaDNA as a quantitative climate proxy. The flowchart depicts sedaDNA transport pathways (river inflow, runoff) into the lake and subsequent steps of data collection, including sediment sampling, laboratory analysis, and bioinformatics processing of sedimentary DNA from lake-surface samples and sediment cores. The**
**calibration step leverages either modern plant occurrence data from GBIF database via a probability density function approach or modern DNA assemblages to set up a WA-PLS regression or a modern analogue matching approach, ultimately enabling reconstruction of the target climate variable (e.g., temperature).**



We used plant assemblages derived from sedimentary ancient DNA (sedaDNA) metabarcoding for quantitative climate reconstructions, applying four different calibration techniques (Fig. 1, Table 1). First, modern plant occurrence data from the

Global Biodiversity Information Facility (GBIF; GBIF.org, 2025) were integrated with contemporary climate data using species distribution modeling (SDM). Model output was used to fit probability density functions (PDFs) used for reconstruction in an indicator-species approach (plantDNA_PDF/SDM). The second approach is similar to the first one, only that GBIF-derived modern taxa occurrences were directly linked to climate data. Third, we directly linked modern sub-fossil sedaDNA assemblages with contemporary climate parameters via Weighted-Averaging Partial Least Squares regression (WA-

PLS) and applied these relationships to assemblages from the past (plantDNA_WAPLS). Finally, modern sub-fossil sedaDNA assemblages were matched to fossil samples via distance measures (plantDNA_MAT). The sedaDNA-based climate reconstructions were further validated through comparisons with pollen-based reconstructions.

## 2.1 Datasets

We used plant DNA metabarcoding data from surface sediment samples (Stoof-Leichsenring et al., 2020; Li et al., 2021). We

subsampled the modern dataset by including only samples from Siberia north of 55°N and filtered for the most abundant taxa (a minimum of 0.075 Hellinger-transformed reads occurring in at least 3 samples) across these surface samples using the chooseTaxa function from the analogue package (Simpson et al., 2025) in the R environment (R Core Team, 2022). Our surface DNA metabarcoding subset yields 203 samples containing a total of 169 taxa.

We applied climate reconstruction to a sedaDNA plant metabarcoding dataset from Lake Billyakh (126.74°E, 65.27°N; 340

m above sea level (a.s.l.) in the western Verkhoyansk Mountains, Siberia (Courtin et al., 2025), which included 59 samples containing 272 Amplicon Sequence Variants (ASVs), representing 256 taxa. To reduce noise in the reconstruction, we discarded samples containing fewer than 20 taxa. Subsequently, using the chooseTaxa function from the analogue R-package, only taxa were retained, which occur in at least 3 samples with a minimum of 0.075 Hellinger-transformed reads, resulting in a final fossil dataset comprising 42 samples and 75 taxa.

A circum-arctic modern plant occurrence dataset derived from GBIF (Lisovski and Schwenkler, 2025) was subsetted to a region spanning 50–220°E and 55–90°N. Courtin et al. (2025) developed a customized plant DNA metabarcoding reference database (*"SibAla_2023"*) specifically for this region. This database encompasses a total of 3398 species, 947 genera, and 223 families, which were consolidated into 2371 ASVs. For constructing the database, GBIF occurrence records were filtered to include only Streptophyta occurrences identified to the species level with at least 10 observations. Species names were

standardized according to the NCBI taxonomy (Courtin et al., 2025) and linked to ASVs derived from the ArcBorBryo (Sønstebø et al., 2010; Willerslev et al., 2014; Soininen et al., 2015), PhyloNorway (Alsos et al., 2022), and EMBL 143 (Kanz et al., 2004) databases. In these cases, when a species corresponds to different ASVs, the ASVs were merged at the species level. All other ASVs were retained individually, named according to the lowest taxonomic level and differentiated by appending a number as a suffix. Consequently, the final customized plant DNA metabarcoding reference database

(*"SibAla_2023"*) consists of 781 unique scientific names (hereafter referred to as taxa).



Mean temperatures of the warmest quarter ("bio10"; hereafter referred to as summer temperatures) and annual precipitation ("bio12") were obtained from the WorldClim 2 dataset (Fick and Hijmans, 2017). WorldClim 2 provides spatially interpolated gridded climate data aggregated from weather stations as temporal averages between 1970 and 2000 and available as GeoTiff files with various spatial resolutions (Fick and Hijmans, 2017). We used data with a spatial resolution of 30 seconds (~1 km$^2$).

**2.2 Methods for sedaDNA calibration**

**2.2.1 plantDNA_PDF/SDM and plantDNA_PDF/GBIF**

We used our customized dataset of modern presence-only plant occurrence from GBIF for calibration. We found that several taxa do not equally cover the temperature range. We therefore used species distribution modeling (SDM) to enhance the point density in the occurrence data and to improve the representation in the climate space as it also includes precipitation in addition

to temperature. SDM was performed with the MaxEnt model (Phillips et al., 2006) via the maxent function in the R-package dismo (Hijmans et al., 2024). MaxEnt is a machine learning-based ecological niche model that is best used in occurrence data-rich situations (>50 data points; Timmermann et al., 2024). Before modeling, we removed the GBIF records that (i) cannot be identified to species level; (ii) originated from fossil specimens; (iii) had a coordinate resolution of lower than 5 km; and (iv) were collected before the year 1981. The GBIF occurrence points for species belonging to the same taxon were merged and

spatially thinned to 15 km. Modern annual precipitation and summer temperature data at a ~1 km² resolution from WorldClim 2 were used as environmental variables to address the major drivers for plant growing conditions. For each taxon, we generated background (pseudoabsence) points with four steps: (i) created a buffer of 200 km distance for all GBIF occurrence points; (ii) extracted the occurrence points of species significantly within the buffer but not belonging to the targeted taxon; (iii) selected the occurrence points that differ from the target taxon in modern climatic conditions as potential background points using an

OCSVM classifier (Senay et al., 2013) via the svm function in the R-package e1071 (Meyer et al., 2024); and (iv) randomly sampled the same number of background points as there are occurrence points of the target taxon. To obtain the optimized combination of features and the regularization multiplier value for the MaxEnt model of each taxon, the model was tuned with the ENMevaluate function in the R-package ENMeval (Kass et al., 2025), which tests regularization multiplier values ranging from 1.0 to 5.0 in increments of 1 and six different feature combinations (L, LQ, H, LQH, LQHP, LQHPT; where L=linear,

Q=quadratic, H=hinge, P=product, and T=threshold) with 5-fold cross-validation, and selects the best results based on the lowest Akaike information criterion (AIC; Warren and Seifert, 2011) value. We also tested regularization multiplier values in increments of 0.5 and more feature combinations, and the prediction results did not change significantly. The prediction was repeated five times for each taxon, and the mean values of suitability and each threshold were used for downstream analyses. The MaxEnt results were binarized by implementing the logistic threshold of 10 percentile training presence. We also tested

other stricter thresholds such as "maximum sensitivity plus specificity (MSS)" as suggested by previous studies (Liu et al., 2016) for several dominant taxa, and found that large areas with a high number of GBIF occurrence points were not included in the binarized ranges. In addition, since absence data are unavailable in our study, specificity estimated from background



data are statistically biased and unreliable (Soley-Guardia et al., 2024). In order to validate our SDM runs, we calculated the Boyce index (Hirzel et al., 2006) from the binarized MaxEnt results. The Boyce Index (-1 to 1) is well suited for presence-

only data, as it estimates the correlation of the predicted abundances with the occurrences. For the 180 taxa that we used for reconstruction in this study 175 showed positive values, that is, models perform better than random, with a median of 0.86 indicating a very good fit for most models. For each taxon, we sampled 500,000 grid boxes from the predicted distribution and extracted summer temperatures from WorldClim 2 based on centroid coordinates of the respective grid cells. We prepared customized distribution datasets with the modeled occurrences for estimating the taxa-temperature relationships.

Taxon-specific climate response curves were estimated by fitting univariate PDFs (Kühl et al., 2002; Chevalier, 2022) to a summer temperature gradient within the study area (i.e. the *"SibAla_2023"*-region). We kept the default threshold of 20 grid cells predefined in the crest.set.modern.data function in the R-package crestr (Chevalier, 2025) with modeled occurrences to estimate PDFs. In the fossil dataset the occurrences of *Pedicularis elata* and *Pulsatilla dahurica* were insufficient to fit a PDF and therefore discarded, resulting in 73 taxa that were used to fit PDFs. With respect to the surface dataset, we discarded 7

taxa which had irregular occurrences and caused unrealistic strong warming as inferred from the assessment of leave-one-out (LOO) analyses of an initial run, leaving 162 taxa for PDF fitting. To ensure a spatially balanced coverage of the climate space in our dataset, we defined a site-specific range of ±7.5°C around the modern observed summer temperature at the reconstruction sites and sampled 100,000 modeled occurrences from each taxon. The site-specific temperature range was further subdivided into equal bins of 1°C width. Within each bin, we either used all available modeled occurrences if the

cumulative number was lower than 100,000 occurrences, or we restricted the total number of all modeled occurrences within the bin by sampling whenever this threshold was exceeded.

We used the crestr R-package (Chevalier, 2022; Chevalier, 2025) to estimate individual PDFs for each taxon and perform summer temperature reconstructions. We employed a customized calibration dataset in the CREST approach which required the preparation of several specific input files including sedaDNA metabarcoding data from Lake Billyakh, modern taxa

occurrence distributions (in our case the samples output from SDMs) alongside a climate space table containing temperature values for each occurrence linked to geographic coordinates, and a taxon table specifying which taxa to include or exclude from the reconstruction. Climate reconstruction using the CREST approach involves three main steps, starting with the calibration of input data, followed by estimating individual climate responses (PDFs) for each taxon, and concluding with the execution of the reconstruction itself. Climate responses for each taxon are estimated individually by fitting a PDF to their

respective temperature ranges. The final temperature reconstruction integrates the PDFs of all taxa selected for inclusion. The PDFs were fitted with 5000 points and a 0.5°C bin width using the crest.calibrate function in the crestr R-package using default setting. The reconstruction was carried out sample-wise by multiplying all PDFs for the selected taxa present in the respective sample and then applied to plant metabarcoding spectra obtained from sediments weighted by their abundance. For details of the CREST reconstruction framework, we refer to Chevalier (2022). For each sample, we obtained the reconstructed optimum

(i.e. the peak in the multiplied PDFs) and the reconstructed mean (i.e. half of the range in the multiplied PDFs) alongside the 50% and 95% uncertainty ranges. Several diagnostic tools from the crestr R-package were used to assess the quality of our





reconstruction. These tools include graphical overviews of climate sensitivities for taxa, plots depicting individual sample PDFs and their reconstruction outcomes, and a leave-one-out (LOO) analysis. The LOO analysis assesses the influence of individual taxa by systematically excluding them from the reconstruction and evaluating the resulting changes. This helps

identify potential biases introduced by taxa indicative of either colder or warmer temperatures (Chevalier, 2022).

The reconstruction using the plantDNA_PDF/GBIF method was carried out in the same way as with the plantDNA_PDF/SDM framework, except that we directly used the original GBIF modern plant occurrences instead of the SDM-sampled occurrences. We extracted summer temperatures from WorldClim 2 for all occurrences in the GBIF dataset and subsetted it to a site-specific range of ±7.5°C around the modern observed summer temperature. PDFs were fitted for the same taxa (73 taxa for Lake Billyakh;

162 taxa for the surface samples) as with the plantDNA_PDF/SDM method.

### 2.2.2 plantDNA_WAPLS and plantDNA_MAT

We used the modern surface samples as a training dataset to perform a reconstruction from the Lake Billyakh sedaDNA metabarcoding data with WA-PLS (ter Braak and Juggins, 1993). We first determined a set of 152 shared taxa in both datasets that are present in at least 10 surface samples. We then restricted both datasets to those taxa. The taxa reads in both datasets

were first Hellinger- and then square-root transformed. Summer temperatures from WorldClim 2 were assigned to each of the 203 surface samples. The reconstruction was carried out with the WAPLS function in the rioja R-package (Juggins, 2024). Prediction errors and model complexity (i.e. the number of components) were estimated by cross-validation. To test the significance of the cross-validated components a randomisation t-test was performed. On this basis we chose component 2 for the model complexity. The standard error of prediction was estimated with bootstrapping.

We performed a reconstruction using MAT (Overpeck et al., 1985). We used the same fossil and modern datasets as for the reconstruction with WA-PLS, but using just Hellinger-transformed reads. The reconstruction was performed with the MAT function in the rioja R-package, in which a model of the 7 closest analogs for the modern data was generated using a square-rooted chord distance (*"sq.chord"*) dissimilarity coefficient. Reconstructed temperatures and bootstrapped standard error of prediction were obtained as weighted means.

### 235 2.2.3 pollen_WAPLS and pollen_MAT

For validation of the sedaDNA-based reconstruction, we performed summer temperature reconstructions on a pollen record (Müller et al., 2010) derived from the same sediment core from Lake Billyakh. We used 194 samples from the northern hemispheric modern pollen dataset presented in Herzschuh et al. (2023b) within 1000 km around Lake Billyakh as training data which equals the range in the modern sedaDNA dataset. Pollen taxa in both, the fossil pollen dataset and the modern

pollen assemblages were harmonized following the criteria described in Herzschuh et al. (2022), i.e. woody taxa and a few very common taxa such as *Artemisia* and *Rumex* were harmonized at the genus level, while herbaceous taxa were collapsed to the family level. Summer temperatures extracted from WorldClim 2 were assigned to each modern sample. Reconstruction methodology follows Herzschuh et al. (2023b). Pollen counts were converted into percentages and square-root transformed.





We used the WAPLS function in the rioja R-package to perform the reconstruction. Informed by results from the cross-
validation and randomisation t-test, we decided to use results from component 1.

We used the same harmonized fossil and modern pollen percentage datasets as with WA-PLS to perform the reconstruction
with MAT via the MAT function in the rioja R-package. Similar to the plantDNA_MAT approach, we used a square-rooted
chord distance dissimilarity looking for 7 close matches in the modern data.

### 2.2.4 Software support

Composite figures were assembled from R outputs using CorelDRAW Graphics Suite 2024. Portions of the manuscript text
were polished with assistance from ChatGPT (OpenAI GPT-4), accessed May 2025.

## 3 Results

We applied four statistical approaches – plantDNA_PDF/SDM, plantDNA_PDF/GBIF, plantDNA_WAPLS and
plantDNA_MAT – to reconstruct summer temperatures using plant DNA metabarcoding spectra obtained from 203 lake
surface-sediment samples collected across Siberia (79-178°E, 55-74°N). These reconstructed temperatures were compared
with observed temperatures to evaluate reconstruction biases.

The plantDNA_PDF/SDM method, calibrated using modern plant occurrence data supplemented by species distribution
modeling, yields a median reconstruction bias of 1.1°C across all surface samples (Fig. 2), representing 9.5% of the total
temperature range (5.0-16.6°C) covered by the dataset (Table 1). Reconstruction accuracy is high at 47.3% of the sites,
exhibiting biases below 1°C. Observed and reconstructed temperatures are strongly correlated (r = 0.86, p < 0.01). The method
generally overestimates temperatures below 8°C and underestimates temperatures above 14°C (Fig. 2). LOO analysis identifies
several taxa – *Larix*, *Persicaria amphibia and Menyanthes trifoliata* and, to a lesser extent, several Ericaceae taxa – as
contributing strongly to warmer reconstructed temperatures confirmed by their sensitivity ranges (Supplementary Fig. A1 and
A2). *Betula*, Anthemideae_01, and Asteraceae_01 have warm effects on the more northern samples, but cold effects on the
more southern samples (Supplementary Fig. A1). Taxa contributing to colder reconstructions include Saliceae, *Dryas*, and
*Hulteniella integrifolia*. Additionally, the plantDNA_PDF/SDM method accurately reconstructed modern temperatures around
Lake Billyakh, as indicated by a small reconstruction bias of only 0.4°C (Table 1).

Compared with plantDNA_PDF/SDM, the plantDNA_PDF/GBIF method, which relies solely on modern plant occurrences
for calibration, shows a substantially higher median bias of 2.1°C (Fig. 2), corresponding to 18.1% of the total temperature
range (Table 1). This larger bias primarily results from pronounced reconstruction inaccuracies at the lower end of the
temperature gradient. In contrast, methods calibrated using modern sediment samples – plantDNA_WAPLS and
plantDNA_MAT – yield relatively low median biases (0.7°C and 0.5°C, respectively; Fig. 2), as determined by the LOO
analysis. However, these methods exhibit significantly lower accuracy when reconstructing the modern temperature around




Lake Billyakh, as indicated by notably higher biases of 3.1°C for plantDNA_WAPLS and 3.8°C for plantDNA_MAT (Table 1).

We applied the four approaches to reconstruct summer temperatures using 73 plant metabarcoding spectra from the Lake Billyakh sediment core covering the last 32,000 years (Fig. 3, 4 and 5). With respect to plantDNA_PDF/SDM, reconstruction uncertainties, inferred from combined PDFs, range from 1.3°C to 1.9°C for the 50% uncertainty range (Fig. 5) and from 3.7°C to 5.4°C for the 95% uncertainty range (not shown). During marine isotope stage (MIS) 3 and 2, summer temperatures were approximately 3.4 ± 1.5°C cooler than present, with a distinct short-term warming peak occurring around 28 ka. At the Last Glacial Maximum (LGM), summer temperatures averaged 10.3 ± 1.6°C. A LOO analysis (Fig. 3) reveals the taxa that have cold and warm effects on the temperature reconstruction in the single samples. Taxa such as *Bistorta vivipara*, *Dryas*, *Eritrichium*, *Lagotis glauca*, *Papaver,* and *Saxifraga oppositifolia* are mainly found in MIS 3 and LGM samples and have cold effects on the reconstruction. The diagnostic plots of the "combinedPDFs" show that most taxa strongly overlap in the cold range for the samples in the glacial (Fig. 4). However, some taxa occur with high abundance that occur today under warm conditions including Crepinidae_01 and Asteraceae_03. Several taxa that occur under intermediate warm conditions, have warm effects during the cold period of the glacial but cold effects during the warm period of the Holocene including Saliceae, Vaccinium_01, and Ranunculus_01 (Fig. 3). The temperature increased during the Late Glacial. The temperature rise accelerated during the Bølling-Allerød interstadial and reached modern-day temperatures around 10.5 ka. Peak warmth occurred in the early mid-Holocene at 5.8 ka, when temperatures were 1.6 ± 1.3°C warmer than present. Following the Holocene Thermal Maximum (HTM), temperatures show a steady decline in the reconstruction (Fig. 4). The total temperature difference between the LGM and HTM at Lake Billyakh is 5.5°C. The diagnostic plots of the "combinedPDFs" for an early Holocene sample (Fig. 4) strongly overlap in the warm range. Compared with the diagnostic plot of the glacial sample, less outlier taxa are observed. *Alnus*, Persicaria_01, and aquatic taxa like Nymphaeaceae and *Potamogeton perfoliatus* became abundant during the Holocene and have warm effects (Fig. 3).

Reconstructed temperature change of the other three methods shows glacial to Holocene trends (Fig. 4) similar to plantDNA_PDF/GBIF. Reconstructions of plantDNA_WAPLS and plantDNA_MAT have an overall larger reconstruction range and show more pronounced late Holocene cooling while the range and cooling intensity is lower for plantDNA_PDF/GBIF compared with plantDNA_PDF/SDM. plantDNA_WAPLS is the only method that does not show relatively warm MIS 3 samples.

We performed pollen-based reconstruction for the same sediment core as with sedaDNA-based reconstruction. Median biases and root-mean squared error of prediction (RMSEP) when comparing to similar calibration methods using sedaDNA are generally higher (Table 1). However, the variations in summer temperatures derived from sedaDNA- and pollen-based reconstruction are in good agreement, although reconstructed temperatures from sedaDNA are overall slightly warmer than from pollen (Fig. 4). The reconstructed temperature range is similar between sedaDNA and pollen_WAPLS and larger with pollen_MAT. The temperature increase during the Bølling-Allerød interstadial is more pronounced in the reconstruction with pollen_MAT compared to the sedaDNA-based reconstruction, while the temperature increase in the reconstruction with



pollen_WAPLS is overprinted by strong variations. The HTM in the pollen-based reconstructions occurs slightly earlier but also during the early mid-Holocene like the sedaDNA-based reconstruction.



**Figure 2: (a) observed vs reconstructed summer temperature and (b) spatial pattern of biases from applying the four calibration approaches to plantDNA spectra from 102 surface sediment samples from eastern Siberia.**




**Table 1: Overview of calibration approaches and performance metrics for plant sedimentary ancient DNA (sedaDNA) and pollen-based temperature reconstructions. For each method, the table gives the median bias (reconstructed – observed) on modern surface samples, modern RMSEP (root-mean-squared error of prediction), calibration temperature range, RMSEP as a percentage of that range; and bias in the Lake Billyakh core-top sample.**

| proxy | description of calibration method | median bias (°C) | RMSEP modern | modern temp. range (°C) | RMSEP % of temp. gradient | core-top bias at Billyakh (°C) |
|---|---|---|---|---|---|---|
| plantDNA_PDF/SDM | Leverages GBIF occurrence data • Simulates modern plant distributions across climate gradients using SDM • Establishes PDFs for plant DNA taxa • Applies PDFs to reconstruct temperatures from 203 surface samples and sediment core | 1.1 | 1.9 | 5.0-16.6 | 16.1 | 0.4 |
| plantDNA_PDF/GBIF | Leverages GBIF occurrence data • Establishes PDFs for plant DNA taxa • Applies PDFs to reconstruct temperatures from 203 surface samples and sediment core | 2.1 | 2.3 | 5.0-16.6 | 19.7 | 0.5 |
| plantDNA_WAPLS | Establishes transfer functions between plant DNA from 203 surface samples and modern climate using WAPLS • Applies transfer functions to plant DNA from surface samples (leave-one-out approach) and Lake Billyakh sediment core | 0.7 | 1.1 | 5.0-16.6 | 9.9 | 3.1 |
| plantDNA_MAT | applies modern analogue technique which matches modern plant DNA spectra from surface samples among each other (leave-one-out approach) and to plant DNA from sediment core • infer modern climate from 7 best analogues as reconstruction result | 0.5 | 1.2 | 5.0-16.6 | 10.5 | 3.8 |
| pollen_WAPLS | Establishes transfer functions between pollen between modern pollen spectra from 194 surface samples from with 1000 km radius around Lake Billyakh and modern climate using WAPLS • Applies transfer functions to plant DNA from surface samples (leave-one-out approach) and Lake Billyakh sediment core | 1.5 | 2.2 | 3.9-16.0 | 18.2 | 1.9 |
| pollen_MAT | Applies modern analogue technique which matches modern pollen spectra from 194 surface samples from with 1000 km radius around Lake Billyakh among each other (leave-one-out approach) and to pollen spectra from Lake Billyakh sediment core • Infers modern climate from 7 best analogues as reconstruction result | 0.6 | 1.6 | 3.9-16.0 | 13.2 | 0.05 |





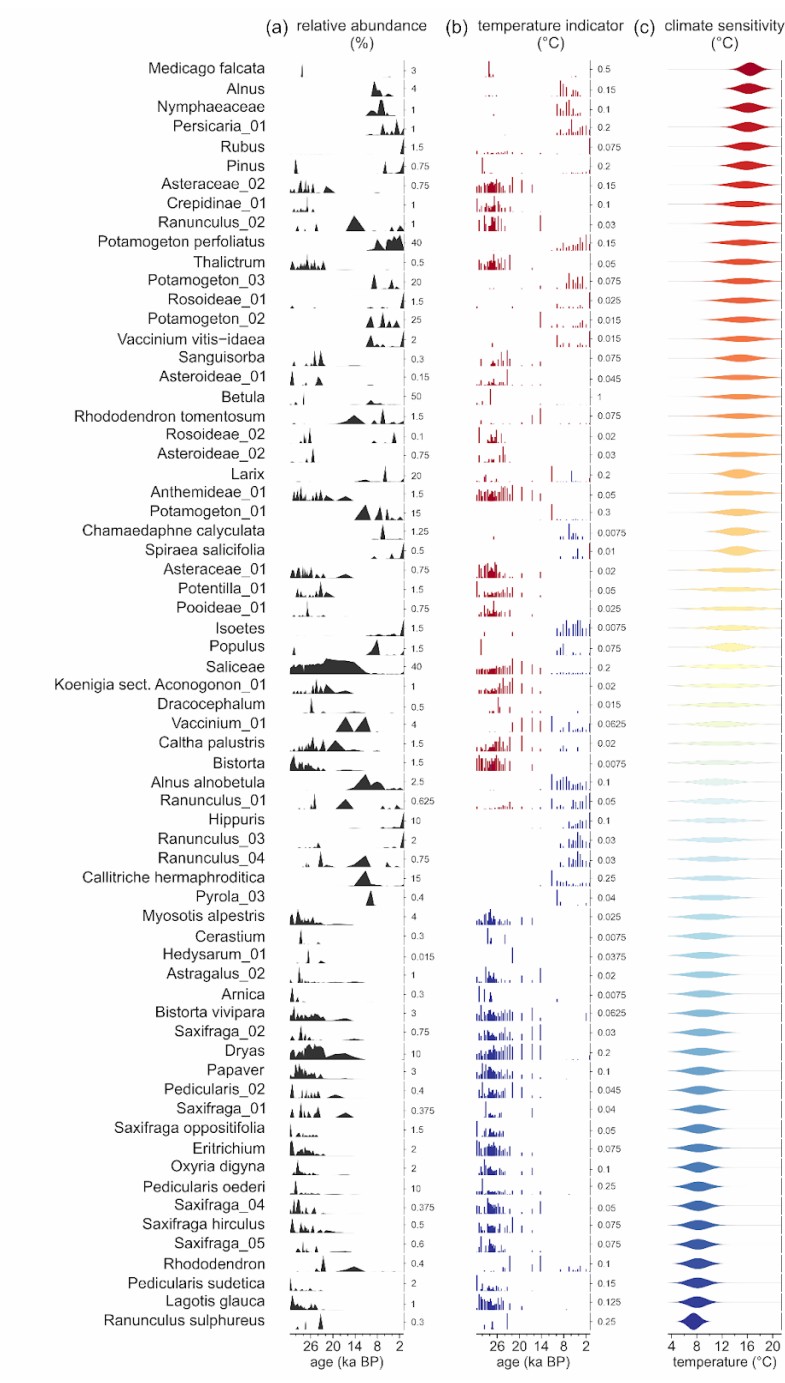

Figure 3: (a) abundances, (b) influence on the reconstruction and (c) sensitivity to temperature of 73 taxa in the Lake Billyakh sedaDNA record used for reconstruction using the plantDNA_SDM/PDF approach, inferred from leave-one-out analyses from the crestr R-package (Chevalier, 2022). The taxa are sorted and color-coded by their temperature optima (i.e. the temperature corresponding to the peak of their PDFs). The height of each bar in the influence plot represents the absolute effect (in °C) of removing the taxon from the reconstruction, and the sign of this effect (increase or decrease of the reconstructed temperature) is color-coded. Here, blue and red bars indicate that the taxon is contributing to a colder or warmer reconstruction.



**Figure 4: (a) summer temperature reconstruction for Lake Billyakh sediment core using the plantDNA_SDM/PDF approach. The light-red read area represents the uncertainties associated with each sample. (b) and (c) Diagnostic "combinedPDFs" plots from the crestr R-package (Chevalier, 2022) showing the combined probability density functions (PDFs) of all taxa recorded in samples dated to 25,818 yr BP and 8,046 yr BP. Line thickness corresponds to the relative weight of each taxon in the sample. The black curve indicates the overall climate reconstruction, with the best climate estimate derived from the maximum of the curve (grey dashed line) and uncertainties assessed from the area under the curve.**




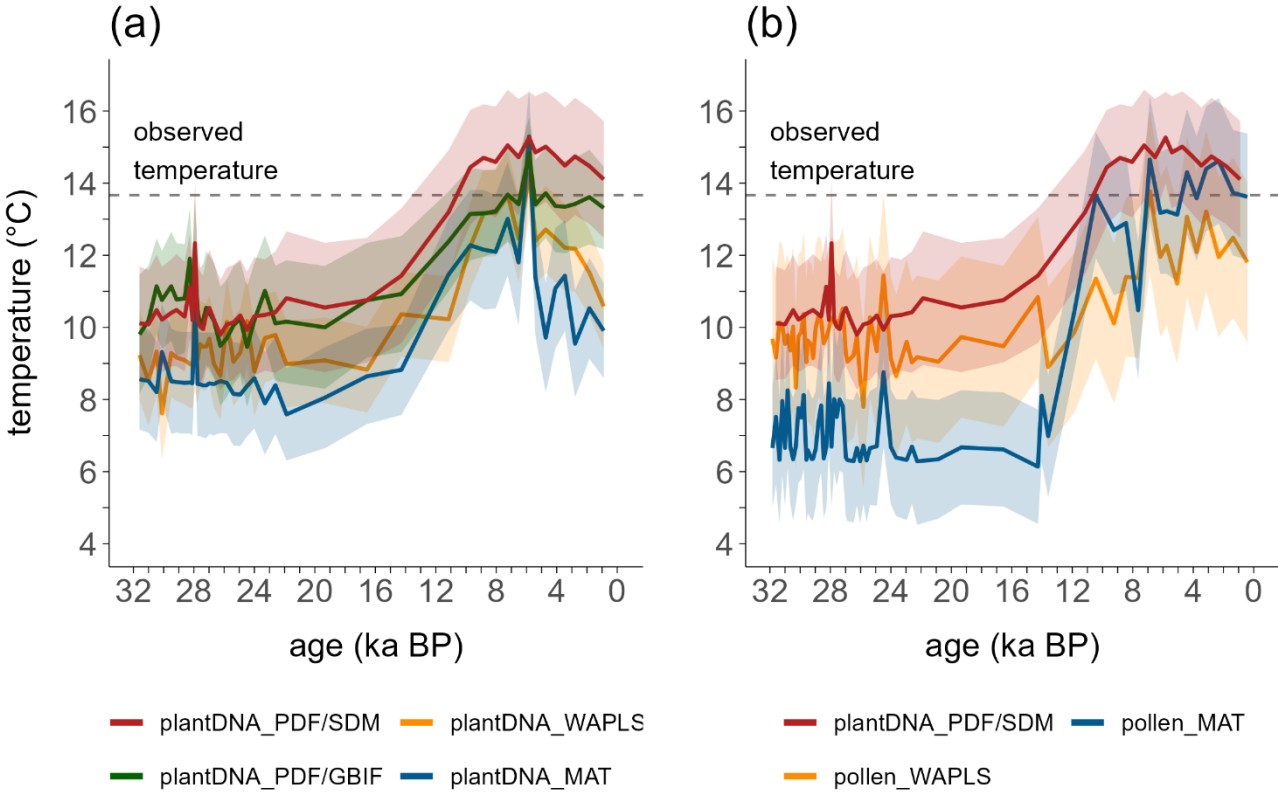

**Figure 5: (a) summer temperature reconstructions for the Lake Billyakh sediment core comparing the results of four reconstruction approaches. (b) summer temperature reconstructions for the Lake Billyakh sediment core comparing plant DNA_SDM/PDF with pollen-based reconstruction results.**

## 4 Discussion

### 4.1 Climate reconstruction leveraging sedimentary ancient DNA

We used sedimentary ancient DNA (sedaDNA) as a novel proxy to quantitatively reconstruct summer temperatures from lake sediment archives. Like other biological proxies commonly used for climate reconstructions, such as pollen (Birks, 2019; Chevalier et al., 2020) and chironomids (Eggermont and Heiri, 2012; Medeiros et al., 2022), surface sedaDNA mirrors the relationship between modern climate conditions and biotic communities.

Validation of our reconstruction approaches using sedaDNA assemblages from 203 surface sediment samples gives a low median reconstruction bias of 0.5°C for calibration based on a modern surface-sample dataset (plantDNA_MAT) and 1.1°C when calibration used species distribution modeling based on modern plant occurrences (plantDNA_PDF/SDM). These median biases and associated RMSEP values from sedaDNA reconstructions (Table 1) are smaller than those typically obtained from other proxies employed in Arctic regions. For example, pollen-climate calibration using WA-PLS from eastern Siberia,



covering a similar temperature gradient, yields an RMSEP of >1.5°C (Herzschuh et al., 2023b, this study, Table 1). Similarly, chironomid-based WA-PLS reconstructions from northeastern Siberia have RMSEP values ranging between 1.5 and 2.5°C

(Nazarova et al., 2015), notably higher than the 1.1°C obtained from our plantDNA_WAPLS model.

Our sedaDNA-derived summer temperature record from Lake Billyakh reveals pronounced variability since the LGM, showing patterns consistent with pollen-based reconstructions from the same site (Fig. 4) and other regional proxy compilations (Biskaborn et al., 2012; Meister et al., 2024). The record indicates strong warming (>5°C) from the last glacial period, with a maximum during the early-to-mid Holocene, followed by a slight cooling towards the late Holocene. These

findings highlight the reliability of sedaDNA as an effective proxy for quantitative long-term temperature trend reconstructions.

We assume that environmental DNA deposited in lake sediments provides a suitable taphonomic context for paleoclimate reconstruction, and in several aspects may be superior to pollen in tracking site-specific climate signals. Lake sediments generally provide stationary proxy signals due to the stable depositional environment (Cohen, 2003). SedaDNA primarily

originates from the lake's immediate catchment, as most DNA is transported via runoff from slopes directly surrounding the lake (Giguet-Covex et al., 2019). Thus, sedaDNA more directly reflects local climate conditions compared to pollen, which is predominantly dispersed by wind and integrates signals from broader regional scales (Herzschuh et al., 2022). For example, *Alnus* occurring with a steady presence in the pollen record throughout the MIS 3 and LGM period, does not support its presence before the Bølling-Allerød in the lake catchment (Fig. 4). Furthermore, sedaDNA largely represents vegetative plant

parts, offering a closer link to actual plant biomass compared to pollen grains, which exhibit taxon-specific and highly variable production rates (Wieczorek and Herzschuh, 2020). Accordingly, sedaDNA effectively captures aquatic taxa, such as *Potamogeton* in the Lake Billyakh record, which is scarcely recorded by the pollen signal. Tracing aquatic taxa is particularly advantageous because they exhibit strong temperature-dependent distribution patterns, especially in subarctic environments where their northern range limits are constrained by summer temperatures (Stoof-Leichsenring et al., 2022). However, using

aquatic taxa for calibration may introduce localized microclimatic effects and impacts from lake-water temperature, potentially biasing the climate signal captured by sedaDNA.

Applying sedaDNA-based reconstructions to sedimentary archives from northern or alpine lakes of glacial origin appears particularly promising. These lakes often contain DNA that is well-preserved due to the cold conditions, neutral pH, and high mineral content (Jia et al., 2022). Nonetheless, low DNA concentrations resulting from dilution effects during periods of high

sedimentation associated with glacier meltwater input may lead to non-representative spectra (Courtin et al., 2021) and to increased susceptibility to contamination during DNA amplification steps, which is commonly indicated by very low richness (Jia et al., 2022). Accordingly, we excluded spectra with <10 taxa from the analyses to stabilize the signal-to-noise ratio of the reconstruction.

Like all vegetation proxies, sedaDNA-based climate reconstructions are subject to several calibration biases, including

potential time lags in vegetation response to climate change (Dallmeyer et al., 2022) confounding effects of multiple climate





drivers on community composition (Chevalier et al., 2020), and complex seasonal biases in biological signals (Bova et al., 2021).

## 4.2 Taxonomic precision in plant metabarcoding strengthens climate signal detection

We employed plant DNA metabarcoding data (Li et al., 2021; Courtin et al., 2025) generated with chloroplast *trn*L *g* and *h*
primers, which are among the most commonly used primers for targeting plants in sedaDNA studies (Taberlet et al., 2007). Our sedaDNA-based reconstruction addresses several limitations associated with traditional plant macrofossil or pollen-based climate reconstructions by enhancing taxonomic resolution and capturing a broader range of taxa, including ferns, mosses, and both terrestrial and aquatic spermatophytes (Capo et al., 2021; Alsos et al., 2024). Within our dataset, 77.2% of identified taxa were resolved to at least genus level, consistent with previously reported results for similar sedaDNA approaches (Alsos et al.,
2018; Sønstebø et al., 2010). We find that taxa identified at the species level typically exhibit narrower sensitivities to climate and (Fig. 6) have such more distinct PDFs, reflecting their more specialized ecological niches and translating into clearer climate signals. In contrast, taxa identified only at the family level, which typically encompass multiple species with broader ecological tolerances, produce PDFs spanning wider temperature ranges, thereby introducing greater noise into climate reconstructions. Consequently, the enhanced taxonomic resolution provided by sedaDNA likely improves our ability to infer
climate conditions from assemblages, as evidenced by the lower reconstruction bias of sedaDNA-based results compared to pollen-based reconstructions when assessed against observed climate data (Fig. 2). The plant metabarcoding assemblages from Lake Billyakh used for reconstruction are substantially more diverse than the pollen data (73 vs. 41 taxa), likely contributing to a more stable reconstruction. In contrast, the pollen-based reconstruction displays greater variability (Fig. 4), possibly due to a lower signal-to-noise ratio stemming from fewer taxa contributing to the reconstruction. This interpretation aligns with
findings by Heiri and Lotter (2010), who demonstrated that lower taxonomic resolution in chironomid-based temperature reconstructions decreases sensitivity in detecting subtle climate variations.

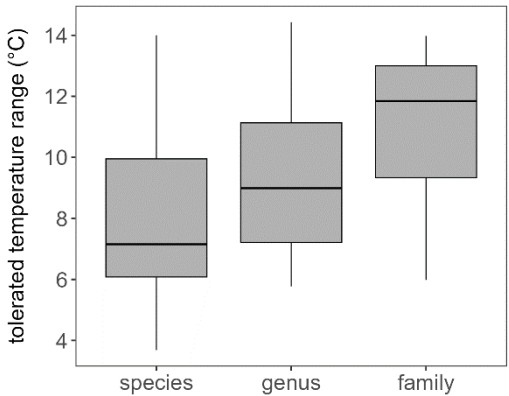

**Figure 6: boxplots of summer temperature 95% uncertainty ranges (as obtained from PDF assessment) of taxa included in plantDNA_PDF/SDM reconstruction. It indicates that the tolerated temperature range increases with lower taxonomic level, i.e.**
**resulting in a reconstruction with a higher uncertainty.**



Examining the joint PDF plots derived from plantDNA_PDF/SDM data from glacial samples reveals outlier taxa, including Crepinidae_01 and Asteraceae_03 (Fig. 5). These taxa consistently occurred under cold glacial conditions despite their modern database records suggesting warm ecological niches. Both sequences belong to the Asteraceae family, known for the limited resolution capability of the *trn*L marker at high taxonomic levels (Sønstebø et al., 2010). This discrepancy likely indicates

hidden taxonomic turnover within these sequence types in Siberia in the past, resulting in their incorrect climate indication in the glacial samples. Additionally, incomplete genetic and floristic databases concerning modern and extinct taxa can also lead to misleading climate reconstruction results (Courtin et al., 2025), emphasizing the importance of careful assessment and validation of sedaDNA reconstructions using assessment tools such as those provided in the crestr R-package.

### 4.3 Reconstruction framework: calibration with sediment surface data vs taxa occurrence databases

We established a novel framework, plantDNA_PDF/SDM, leveraging species distribution modeling to enrich observation data of modern plant occurrences obtained from GBIF (GBIF.org, 2025). We linked these simulated taxa occurrences – resolved at the genetic-marker level – to contemporary climate conditions using probability density functions (Kühl et al., 2002), as implemented in the CREST framework (Chevalier et al., 2014). These PDFs were then applied to extract climate signals from past plant communities inferred through sedimentary ancient DNA. Overall, the plantDNA_PDF/SDM framework

demonstrated superior performance with our data compared to traditional frameworks typically employed for assemblage-based climate proxies. PlantDNA_PDF/SDM shows a low median bias of 1.1°C for the modern sediment surface samples when comparing the reconstruction with observation; this is much lower than for PlantDNA_PDF/GBIF, which yielded a median bias of 2.1°C. The RMSEP of plantDNA_PDF/SDM is roughly similar to leveraging traditional calibration approaches, which yield 1.1°C for plantDNA_WAPLS and 1.2°C for plantDNA_MAT (Fig. 2). However, in contrast to the PDF-based

approaches, they yield a high bias for the Lake Billyakh core top reconstruction of >3°C (Table 1).

As with all proxy methods that infer climate from taxonomic assemblages, our sedaDNA-based reconstructions display systematic biases at the extremes of the temperature gradient – a phenomenon known as the edge effect (Birks, 1995). Specifically, we find a warming bias at sites characterized by colder temperatures (<8°C), predominantly affecting high-latitude localities, whereas temperatures at warmer sites (>14°C) tend to be underestimated (Fig. 2). Consequently,

reconstructions of colder climates, such as those during the glacial periods, likely represent conservative temperature estimates and might underestimate true temperature variability. The edge-effect is particularly strong for the plantDNA_PDF/GBIF, which likely originates from the poor coverage of the GBIF plant occurrences in northern Siberia (Fig. 2).

A notable advantage of our novel plantDNA_PDF/SDM framework compared with methods that require modern calibration sets from surface sediments, is its reliance on presence/absence data in the modern occurrence data, significantly improving

spatial coverage for the training dataset. Our reconstructions may still be influenced by spatial biases inherent in the availability of modern plant distribution data. For example, the spatial resolution used to model the taxon-climate relationships (1 km²) could pose limitations, particularly in regions with complex topography, such as mountainous areas. PDF-based reconstructions may, on the other hand, increase the sensitivity-to-noise ratio, particularly from taxa with sparse or marginal



occurrences at sites with special growing conditions. For example, we identified several taxa, such as *Koenigia* sect.
*Aconogonon_02*, Fragariinae, and aquatic plants like Nymphaeaceae, that induce a warming bias, particularly at the northernmost sites (Fig. A1). Fossil samples sporadically containing these taxa show some flickering, so we decided to generally exclude taxa with very low abundances from the analysis to stabilize the signal-to-noise ratio in the reconstructions. As is generally common for all calibrations, our preliminary tests showed some sensitivity to the set temperature range and thus we chose a reasonable range guided by known temperature variability for glacial-interglacial time-scales.

**4.4 Outlook**

The further advancement and broader application of sedaDNA as a quantitative climate proxy can be achieved by addressing several key research areas. Controlled taphonomy and transport experiments are essential for a better understanding of sedaDNA deposition processes, ultimately enabling fully mechanistic proxy-forward models. Additionally, sedaDNA proxy validation using laminated sediment sites would provide detailed insights into temporal signal fidelity by allowing direct
comparisons with observational climate data.

With plant DNA, proxy refinement can be enhanced by incorporating additional barcode markers (e.g., *ITS*2, *rbc*L), employing shotgun sequencing approaches, increasing taxonomic resolution, and extending applicability across a wider range of taxa (Liu et al., 2021). Furthermore, integrating absolute quantification methods (e.g. DNA-based biomass estimates) could reduce PCR-related biases, enhancing proxy accuracy (Ushio et al., 2018).

With respect to calibration, using beyond-classical statistical methods such as machine learning approaches (e.g. random forests) could leverage complex, non-linear relationships within multi-taxon assemblages, offering improved predictive power and adaptability to heterogeneous datasets (Salonen et al., 2019).

The full incorporation of species distribution modeling into PDF fitting could further enhance calibration robustness, particularly in data-sparse regions (Elith and Leathwick, 2009). Additionally, coupling sedaDNA data with trait-based
community metrics (e.g. leaf phenology, woodiness) can offer mechanistic connections to climate variability. Such insights, beyond simple taxonomic approaches, can better address challenges arising from non-analog situations such as historical low atmospheric $CO_2$ during the Last Glacial Maximum (Prentice et al., 2022).

Expanding the applicability of sedaDNA through multi-site studies across diverse archives (e.g. lake sediments, peat, permafrost) will strengthen its utility as a climate proxy. Integrating sedaDNA with complementary plant proxies such as
pollen and macrofossils – for example within hierarchical Bayesian frameworks – can systematically propagate key sources of uncertainty, including taphonomic, taxonomic, calibration, and temporal, into comprehensive probabilistic climate reconstructions.





## 5 Conclusion

We demonstrated the effectiveness of sedimentary ancient DNA (sedaDNA) as a novel biological proxy for quantitative
reconstruction of summer temperatures from lake sediments. This approach was successfully applied to plant DNA
metabarcoding data from surface sediments and a sediment core from eastern Siberia. Calibration was performed either by
linking modern plant occurrences from the GBIF database to summer temperatures using probability density functions, or by
relating modern DNA assemblages directly to climate via weighted averaging partial least squares (WA-PLS) regression or
modern analogue matching techniques. The sedaDNA-based reconstructions consistently provided accurate and robust climate
estimates, exhibiting lower biases and smaller prediction errors compared to traditional proxies such as pollen and chironomids.
The improved performance of sedaDNA-based methods primarily stems from their higher taxonomic resolution, enabling
clearer ecological and climatic interpretations. Furthermore, sedaDNA reflects local vegetation conditions more directly,
owing to its immediate catchment origin, whereas pollen integrates signals over broader, regional scales. However, like all
assemblage-based methods, sedaDNA reconstructions exhibit systematic biases such as the "edge effect", typically
overestimating cooler climates and underestimating warmer climates. Addressing these biases, as well as carefully managing
taxa with sparse occurrences, is essential for further improving the reliability of sedaDNA-based climate reconstructions.






## Appendix A

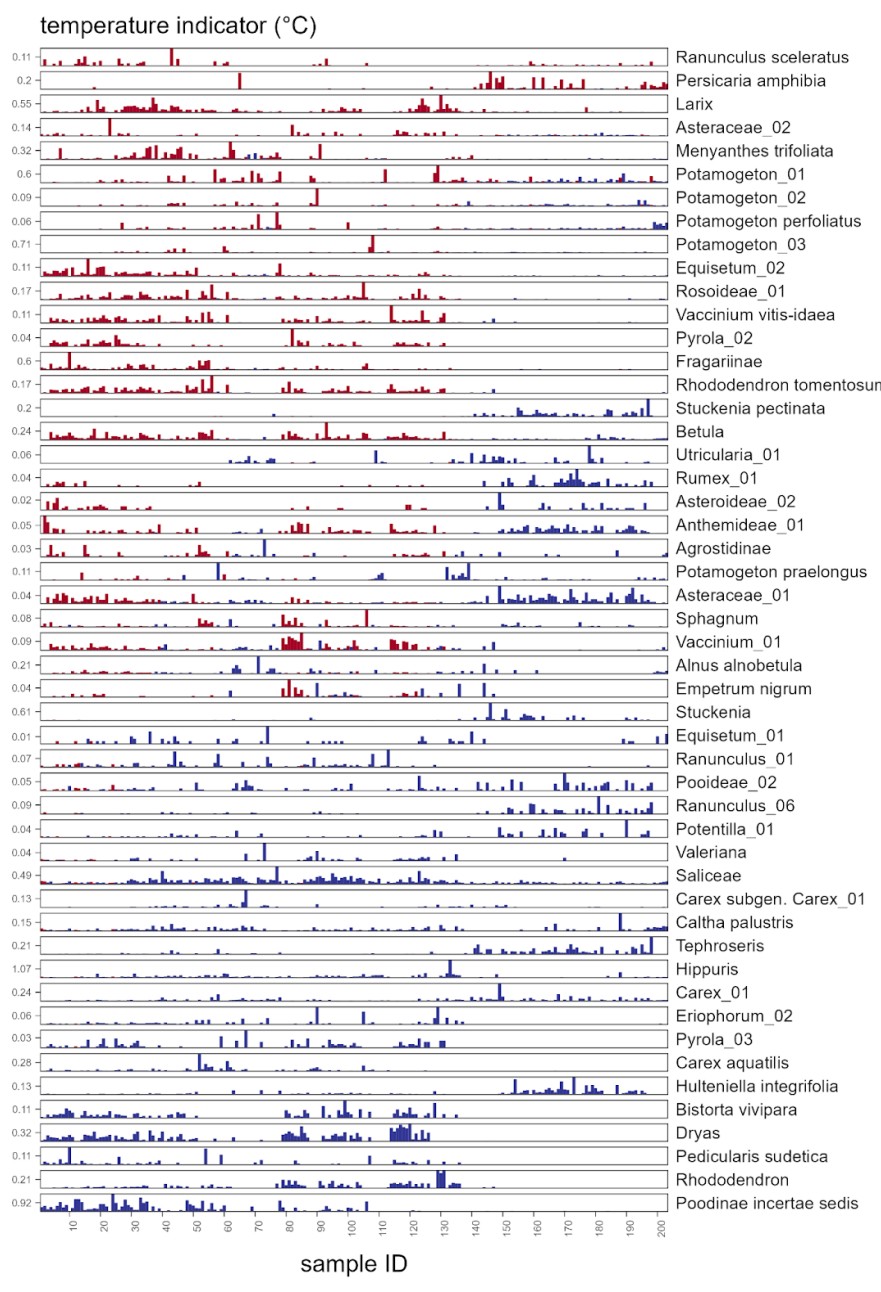


**Figure A1: influence on the summer temperature reconstruction of 50 taxa that occur in the most of the 203 lake-surface sediment samples from Siberia from the plantDNA_SDM/PDF approach inferred from leave-one-out analyses in the crestr R-package (Chevalier, 2022). The taxa are sorted by their temperature optima (i.e. the temperature corresponding to the peak of their PDFs), samples are sorted according to their modern temperatures.**






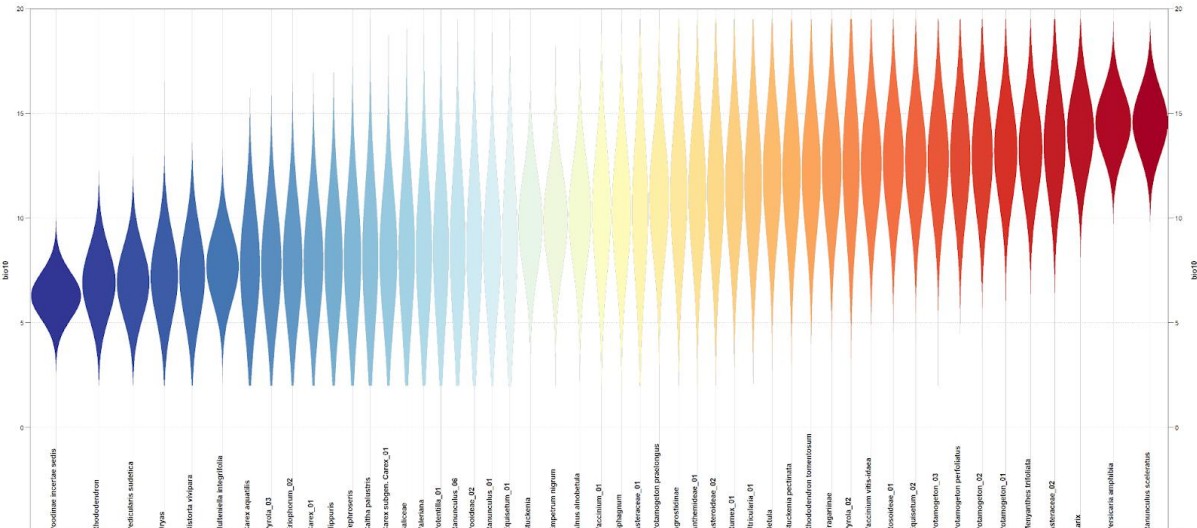

**Figure A2: sensitivity to temperature for 50 taxa that occur in the most of the 203 lake-surface sediment samples from Siberia from the plantDNA_SDM/PDF approach. The taxa are sorted and color-coded by their temperature optima (i.e. the temperature corresponding to the peak of their PDFs).**


## Code and data availability

R-code is provided for the sedaDNA-based reconstruction of mean summer temperatures for Lake Billyakh and for the 203 surface samples across the "*SibAla_2023*"-region, as well as for reproducing the figures shown in this manuscript. The script and the datasets with fossil and surface metabarcoding data alongside their customized distribution and climate space datasets are available on Zenodo: https://doi.org/10.5281/zenodo.15579584. For review purposes, access to the files are restricted. Please contact the authors or the editor for access.

## Author contributions

UH designed the study. The analyses were designed by UH, implemented by TB supervised by UH. UH and TB wrote the initial draft of the manuscript. WJ and SL helped with the implementation of the SDMs. SL created Fig. 1 based on a draft of UH. All co-authors commented on the initial version of the manuscript.

## Competing interests

The authors declare that they have no conflict of interest.



**Acknowledgements**

The modern plant occurrence data were obtained from the Global Biodiversity Information Facility (GBIF). We would like to
express our gratitude to all data contributors, data stewards, and the GBIF community. We thank all the participants in the
field work campaigns who obtained the PG1755 sediment core from Lake Billyakh and collected the surface DNA samples
from across Siberia. We are grateful to the team at AWI in Potsdam that implemented the laboratory work to establish the
sedaDNA dataset. We thank Cathy Jenks for language editing.

**Financial support**

This research has been supported by the European Research Council (ERC Glacial Legacy; grant no. 772852) and by the
German Research Foundation (DFG) Gottfried Wilhelm Leibniz Award (grant no. HE 2622/34-1) to Ulrike Herzschuh. She is
also supported by the German Federal Ministry of Education and Research (BMBF) through the PalMod Phase III project
(grant no. FKZ:01LP2306B).

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
