# Peer review of "Quantitative climate reconstruction from sedimentary ancient DNA: framework, validation and application"

_EGUsphere, 2025_

## Author Comment (AC1)

**Quantitative climate reconstruction from sedimentary ancient DNA: framework, validation and application**

**Response to comments of Anonymous Referee #1**

**Reviewer comment: (1)** Pollen preserved is lake sediments have been used to generate quantitative reconstructions of past climate for the last 50 years, but two inherent problems severely limit the precision possible. First, species with ecologically distinct niches can be difficult or impossible to separate palynologically, secondly, some wind dispersed species produce vast amounts of well dispersed pollen that can blow into lakes far beyond the ecological limits of the species. This manuscript tests the potential of ancient DNA, which has high taxonomic resolution and limited dispersion, as an alternative to pollen for reconstructions. The manuscript tests different reconstruction methods, including traditional transfer functions that use a calibration set and an alternative method that uses presence-only data.

The methods used are:

- CREST based on GBIF occurrence data
- CREST based on GBIF occurrence data with MaxEnt preprocessing
- WAPLS using a calibration set
- MAT using a calibration set.

The first two methods are just applied to the aDNA data, the latter two are applied to both aDNA and pollen data. This may be the first time a head-to-head comparison of CREST and transfer functions has been done. It might be worth extending the analysis to run CREST with the pollen data as well for a more complete comparison.

Our response: We have now implemented the CREST analyses using the standard pipeline using the data sources in the crestr R package. Since this is not the focus and scope of the paper we included the results in the supplements.

New text: For completeness, we also ran CREST on pollen data using CREST's built-in standard pipeline (see Appendix Fig. A3). The pollen-based CREST reconstruction shows similar long-term trends but less variance compared to the sedaDNA-based reconstruction.

New Appendix text:

**Methods:**

Using the CREST's built-in standard pipeline requires a proxy-species equivalency (PSE) table as fossil taxa are most often identified at a lower taxonomic resolution. Hence, all individual species available in the CREST database that are likely to have produced a certain fossil taxon need to be associated with their corresponding fossil taxon (Chevalier, 2022). We therefore determined the taxonomic rank for all harmonized fossil pollen taxa and indicated their respective level (1 = family level, 2 = genus level) alongside the taxonomic lineage in the PSE table. The reconstruction was conducted in the same way as with the plantDNA_PDF/SDM and plantDNA_PDF/GBIF methods, except that we used the proxy-species equivalency table instead of a customized distribution table.

**Results:**

The pollen-based CREST reconstruction shows similar long-term trends but less variance compared to the sedaDNA-based reconstruction. The assessment and potential optimization of the pipeline was not in the focus of this manuscript.

new Appendix Figures:

[Figure]

**Fig. A2:** comparison between sedaDNA- and pollen-based summer temperature reconstruction using the CREST framework.

[Figure]

**Fig. A3:** influence on the summer temperature reconstruction using the CREST framework with harmonized pollen data inferred from leave-one-out analyses in the crestr R-package (Chevalier, 2022).

Remark to the reviewer on Fig. A3: we will revise the Figure and bring it to the same layout and color scheme as e.g. Fig. A1 when uploading the revised manuscript.

**Reviewer comment: (2)** The justification for using CREST with MaxEnt preprocessing is "that several taxa do not equally cover the temperature range". This is a rather vague justification, and I'm not exactly sure what is meant by it. The GBIF-MaxEnt-CREST pipeline is novel and rather involved. I recommend starting this section with a short paragraph that outlines the process so that the details are easier to follow.

Our response: We agree that it can be justified better i.e. revising the text to make clear that SDMs (using the MaxEnt model) help to provide a more extensive model prediction bases - basis for the CREST analysis. We revised the text accordingly.

New text: GBIF occurrences are especially biased in remote northern regions, providing highly valuable observations but limited to certain areas that are more easily accessible. Therefore, we developed species distribution models (SDMs) that predict the potential spatial range of the species and took a random sample within the range to derive a more robust temperature niche estimate (Appendix Fig. A1).

new Appendix Figure:

[Figure]

**Fig. A1:** PDFs fitted to *Alnus* occurrences when using (a) the original GBIF occurrences and (b) modeled occurrences derived from SDM to determine temperature niche estimates.

**Reviewer comment: (3)** Pre-processing with MaxEnt seems to improve the performance of CREST, but I do not understanding what exactly how MaxEnt helps CREST perform better. I'm dubious of the claim that it "enhance[s] the point density in the occurrence data" as it is not possible for the method to create data. Maybe a plot comparing the niches estimated by both methods would help explain what is happening.

Our response: The justification was indeed too vague. MaxEnt predicts continuous distributions across space. In short, the model is fitted by the climate niche (n-dimensional) derived from observations, enhanced by random background samples from regions of unlikely occurrences and predicts the climate suitability for the entire study area. This approach helps to reduce the sampling bias (due to human observed occurrences), fills gaps notably in under-represented parts of the climate gradient, leading to more realistic and robust PDFs. In addition to the better description (first paragraph of 2.2.1 - see also comment above), we added a new Appendix Fig. A1 alongside a description to the Appendix.

New text: GBIF occurrences are especially biased in remote northern regions, providing highly valuable observations but limited to certain areas that are more easily accessible. Therefore, we developed species distribution models (SDMs) that predict the potential spatial range of the species and took a random sample within the range to derive a more robust temperature niche estimate (Appendix Fig. A1). SDMs were implemented using the MaxEnt model (Phillips et al., 2006) from the R-package dismo (Hijmans et al., 2024).

new Appendix Figure:

[Figure]

**Fig. A1:** PDFs fitted to *Alnus* occurrences when using (a) the original GBIF occurrences and (b) modeled occurrences derived from SDM to determine temperature niche estimates.

**Reviewer comment: (4)** I can easily imagine that the CREST is too constrained in the niche shape they can fit, and it might be profitable to allow more than normal or log-normal PDFs. So rather than pre-processing the data with MaxEnt, the first step of CREST is replaced by MaxEnt (or another flexible model). Of course, the penalty for using more flexible models is that they are prone to over-fitting.

Our response: We totally agree, different and maybe even more flexible models or distributions might offer advantages, yet with the potential for over-fitting. We added a sentence in the discussion to make readers aware of potential future developments and pitfalls.

New text: While our pipeline uses MaxEnt as preprocessing for CREST PDFs, an alternative would be to replace the PDF step entirely by a more flexible niche model (e.g. Maxent). This could further improve the niche shapes but may also lead to overfitting.

**Reviewer comment: (5)** There is an issue with the cross-validation of the transfer function models. The ms reports that the uncertainties on the reconstruction are calculated using bootstrapping, but it is unclear what cross-validation scheme is use to estimate the models' performance.

Our response: Added.

New text: Informed by results from the leave-one-out cross-validation and randomisation t-test, we decided to use results from component 1.

**Reviewer comment: (6)** One widely used cross-validation scheme is leave-one-out cross-validation. Somewhat confusingly, this ms, following Chevalier (2022), uses the term leave-one-out to refer to a type of sensitivity analysis in which taxa are left out sequentially. It would be better to call this step a sensitivity analysis.

Our response:  We agree.

New text: These tools include graphical overviews of climate sensitivities for taxa, plots depicting individual sample PDFs and their reconstruction outcomes, and a sensitivity analysis (originally termed leave-one-out analysis in Chevalier (2022)).

**Reviewer comment: (7)** The ms emphasises the median bias of the reconstructed temperatures as a metric of transfer function performance. I have not seen this metric used before. Mean bias is sometimes reported, but not prominently as it can be zero even if the transfer function has no skill. Median bias will have the same unfortunate property. Maximum bias is more useful.

Our response: We actually used median absolute bias. We now clarify this and also report maximum absolute bias in Tab. 1.

New text: We report both median absolute bias and maximum absolute bias to better capture systematic offsets in the reconstructions.

**Reviewer comment: (8)** The pollen-MAT error for the Billyakh core top is very low. Is this lake part of the calibration set? If so, such a low error is not surprising, and it might be worth removing before reconstructing the core top it to get an unbiased estimate. The same applies to the WAPLS model, but the effect will be much weaker and could probably be ignored (unless the assemblage is distinctive), but treat the two transfer function methods the same way.

Our response: Surface samples or core-top samples from Billyakh were not part of the pollen calibration set. We double-checked, it seems that it is just good luck that the transfer function works so well for the Billyakh core top using MAT.

New text: Samples from the lake were not included in the calibration set.

**Reviewer comment: (9)** If Billyakh is one of the calibration set sites, it could be marked on the panels on the left of figure 2.

Our response: Does not apply because it is not part of the calibration set.

**Reviewer comment: (10)** How was it decided to use seven analogues in the MAT models? (seven should be written in words, as should any other small integers).

Our response: We wanted to stay consistent with the approach in Herzschuh et al. (2023b)

New text: We used a square-rooted chord distance dissimilarity looking for seven close matches in the modern data following Herzschuh et al (2023b).

**Reviewer comment: (11)** I don't know if it needs to be stated in the ms, but it may be worth reminding readers that there is a risk of circular reasoning when interpreting assemblage changes due to climate when that assemblage has been used to reconstruct the climate.

Our response: We agree. Added.

New text: Finally, we note the risk of circularity when inferring drivers of vegetation change from the same assemblages used to reconstruct climate; therefore, our reconstructions should not be used to attribute drivers of vegetation change.

---

## Author Comment (AC2)

**Quantitative climate reconstruction from sedimentary ancient DNA: framework, validation and application**

**Response to comments of Referee #2 (Charline Giguet-Covex)**

**Reviewer comment: (1)** This manuscript aims to reconstruct past summer temperatures in Siberia over the last 32,000 years. Four approaches are developed based on modern datasets (observations and fossil records) and applied to a sediment archive (a core from Lake Billyakh). Each method entails specific benefits and drawbacks, which makes the approaches highly complementary and strengthens the robustness and quality of the study. In particular, the approaches combining "raw" GBIF data or species distribution models derived from these data (a method mostly used in ecology) with taxon-specific probability density functions to link taxa to climatic conditions are very original in palaeoenvironmental contexts.

Our response: Thank you for this comment.

**Reviewer comment: (2)** It is also noteworthy that using plant sedaDNA communities in combination with several statistical approaches significantly improved predictive accuracy (see median bias and root-mean-squared error of prediction), compared to other proxies (e.g. pollen, chironomids). Another strength of the study is the availability of a large modern surface-sample dataset (203 sites), which was used for training (WA-PLS, MAT) and for independent validation of the methods (GBIF-SPD or SDM-SPD).

Our response: Thank you for this comment

**Reviewer comment: (3)** My main concern relates to the impact on reconstructions that incorporate the SPDs approach of the relative contributions of different taxa estimated by sedaDNA, which most likely do not reflect their actual contributions in the landscape. Since this information is used to weight the SPDs, I think the potential impact should be discussed. This could be addressed in section 4.3. At present, the only mention is in the outlook section: "Furthermore, integrating absolute quantification methods (e.g. DNA-based biomass estimates) could reduce PCR-related biases, enhancing proxy accuracy (Ushio et al., 2018)."

Our response: We addressed this comment by adding some discussion.

New text: In our PDF-based reconstructions, the joint climate signal is weighted by the relative read abundances of taxa to reduce stochastic noise compared to unweighted presence–absence approaches. However, sedaDNA read counts are influenced, among others, by PCR bias, marker specificity, and taphonomic processes, and thus do not necessarily reflect the true abundance of taxa in the landscape (Elbrecht and Leese, 2015; Giguet-Covex et al., 2019).

**Reviewer comment: (4)** In addition, in my view, the manuscript would benefit from a section addressing other climate proxies beyond vegetation assemblage reconstructions (e.g. BrGDGTs, chironomids, δ18O from diatoms or ostracods).

Our response: We now addressed this in the discussion.

New text: Co-located multi-proxy studies pairing sedaDNA with independent temperature proxies are required, because these proxies differ in seasonality, source area, and taphonomy. Candidates include brGDGT palaeothermometry (De Jonge et al., 2014), chironomid head-capsule transfer functions for lake-summer temperature (Eggermont and Heiri, 2012), and lacustrine $\delta^{18}O$ in carbonates or diatom silica (Leng and Marshall, 2004; Meister et al., 2024).

**Reviewer comment: (5)** Just one more question that I forgot to include in the review: for the WA-PLS you perform 2 transformations (Hellinger- and then square-root transformed) and for the MAT you only do the Hellinger transformation). Can you explain why?

Our response: Actually, it is similar between the two approaches, i.e. we used Hellinger- and square-root transformation. Because in MAT the additional square-root transformation is already included in the distance metrics sq.chord (rioja R package) which we used for implementing the reconstruction.

**Specific comments**

**Reviewer comment: (6)** L27: I would add *"in appropriate contexts, i.e. where vegetation composition is primarily driven by climatic conditions."*

Our response: We did not include it to keep the abstract short. But we explicitly point to this prominently in the third sentence in the introduction.

New text: These approaches build upon the assumption that compositional changes within biotic communities are predominantly driven by changing climatic conditions (Birks et al., 2010).

**Reviewer comment: (7)** L50: Perhaps mention *topography* as well?

Our response: We added "topography" to the list of information that pollen signals integrate over larger spatial scales.

**Reviewer comment: (8)** L285–286: The sentence is awkward: *"However, some taxa occur with high abundance that occur today under warm conditions including Crepinidae_01 and Asteraceae_03."* → Suggested: *"However, some highly abundant taxa also occur today under warm conditions (e.g. Crepinidae_01 and Asteraceae_03)."*

Our response: Thank you, we rephrased the sentence according to your suggestion.

New text: However, some highly abundant taxa also occur today under warm conditions (e.g. Crepinidae_01 and Asteraceae_03).

**Reviewer comment: (9)** L294: For *"outlier taxa"*, please provide a definition: e.g. taxa with warm ecological preferences found during glacial periods, or the opposite.

Our response: We specified "outlier taxa" to "outlier taxa with warm ecological preferences".

**Reviewer comment: (10)** L339: The word *"quantitatively"* seems unnecessary — temperature is inherently quantitative.

Our response: in this context, "quantitatively" refers to reconstruction, not to temperature.

**Reviewer comment: (11)** L349–350: Specify that you are referring to *chironomid head capsules*, not chironomid DNA.

Our response: we changed "chironomid-based WA-PLS reconstructions" to "chironomid-based WA-PLS reconstructions from head capsules"

**Reviewer comment: (12)** L359–361: Runoff refers only to the flow of water over the land surface (rain, snowmelt). Strictly speaking, this does not necessarily include erosion, which is the key process in Giguet-Covex et al. (2019). Suggested formulation: *"SedaDNA mainly originates from the lake's immediate catchment, since erosion of the surrounding slopes is the dominant transport process."*

Our response: thank you, we rephrased the sentence according to your suggestion.

New text: SedaDNA mainly originates from the lake's immediate catchment, since erosion of the surrounding slopes is the dominant transport process (Giguet-Covex et al., 2019).

**Reviewer comment: (13)** L375: With *Courtin et al., 2021*, you may also cite *Giguet-Covex et al., 2019*.

Our response: We added *Giguet-Covex et al., 2019* to the citation.

**Reviewer comment: (14)** L380: I would precise "...of direct and indirect climate drivers..."

Our response: We changed "...of multiple climate drivers on community composition" to "...of direct and indirect climate drivers on community composition".

**Reviewer comment: (15)** L391: Figure 6 is not placed correctly in the sentence.

Our response: thank you for the notice. We placed *(Fig. 6)* correctly now.

**Reviewer comment: (16)** L396–401: The explanation could be clarified. Current text:*"The plant metabarcoding assemblages from Lake Billyakh used for reconstruction are substantially more diverse*

*than the pollen data (73 vs. 41 taxa), likely contributing to a more stable reconstruction. In contrast, the pollen-based reconstruction displays greater variability (Fig. 4), possibly due to a lower signal-to-noise ratio stemming from fewer taxa contributing to the reconstruction. This interpretation aligns with findings by Heiri and Lotter (2010), who demonstrated that lower taxonomic resolution in chironomid-based temperature reconstructions decreases sensitivity in detecting subtle climate variations."*

→ Suggestion: Make the link between diversity and stability explicit. For example:

*"A higher number of taxa increases the likelihood of detecting species with narrow ecological niches, which may strengthen the climate signal and improve the signal-to-noise ratio. This mechanism, rather than diversity per se, may explain the stability of the sedaDNA-based reconstruction."* Also, clarify the distinction between *diversity* and *taxonomic resolution* to avoid confusion (you first speak about diversity and at the end about taxonomic resolution).

Our response: Thank you. We implemented this suggestion

New text: The plant metabarcoding assemblages from Lake Billyakh used for reconstruction are substantially more diverse than the pollen data (73 vs. 41 taxa) likely due to the better taxonomic resolution in the sedaDNA signal. A higher number of taxa increases the likelihood of detecting taxa with narrow ecological niches, which may strengthen the climate signal and improve the signal-to-noise ratio.

**Reviewer comment: (17)** L407: I assume you mean Figure 4.

Our response: yes, Fig. 4 is the correct reference. We corrected it in the text.

**Reviewer comment: (18)** L423: Please provide the RMSEP value for the plantDNA_PDF/SDM approach.

Our response: We added the RMSEP of 1.9°C for plantDNA_PDF/SDM to the text.

**Reviewer comment: (19)** L425–427: Clarify whether you are referring to all approaches or only the PDF-based ones. The results for WA-PLS and MAT in Fig. 2 seem less clear.

Our response: We clarified.

New text: The RMSEP of plantDNA_PDF/SDM (1.9°C) is roughly similar to leveraging traditional calibration approaches using modern proxy-based assemblage datasets, which yield 1.1°C for plantDNA_WAPLS and 1.2°C for plantDNA_MAT (Fig. 2). However, in contrast to the PDF-based approaches, they yield a high bias for the Lake Billyakh core top reconstruction of >3°C (Table 1).

**Figure improvements**

**Reviewer comment: (20)** Fig. 1: Add a blue border to the text boxes associated with sedaDNA (field sampling, laboratory analyses, bioinformatics). In the bioinformatics part, remove animals (since only plant data were used here).

Our response: Thank you. We implemented the suggestions.

**Reviewer comment: (21)** Fig. 2: The ΔT for Billyakh is not shown — could you add its color code? Also, include the number of modern lake sediment sites used.

Our response: We prefer to not indicate this in the figure. This may be misleading as Billyakh is not part of the calibration set.

**Reviewer comment: (22)** Fig. 4: Highlight (e.g. in bold) taxa detected in both periods.

Our response: Thank you. We implemented the suggestions (see below).

[Figure]